# Evaporation-Driven Energy Generation Using an Electrospun Polyacrylonitrile Nanofiber Mat with Different Support Substrates

**DOI:** 10.3390/polym16091180

**Published:** 2024-04-23

**Authors:** Yongbum Kwon, Dai Bui-Vinh, Seung-Hwan Lee, So Hyun Baek, Songhui Lee, Jeungjai Yun, Minwoo Baek, Hyun-Woo Lee, Jaebeom Park, Miri Kim, Minsang Yoo, Bum Sung Kim, Yoseb Song, Handol Lee, Do-Hyun Lee, Da-Woon Jeong

**Affiliations:** 1Korea National Institute of Rare Metals, Korea Institute of Industrial Technology, Incheon 21655, Republic of Korea; kyb916@kitech.re.kr (Y.K.); leesh93@kitech.re.kr (S.-H.L.); qorthgus9@kitech.re.kr (S.H.B.); yjj0011@kitech.re.kr (J.Y.); totptkd12@kitech.re.kr (H.-W.L.); bskim15@kitech.re.kr (B.S.K.); songys88@kitech.re.kr (Y.S.); 2Department of Environmental Engineering, Inha University, Incheon 22212, Republic of Korea; daibv@ntu.edu.vn (D.B.-V.); james803112@naver.com (J.P.); leehd@inha.ac.kr (H.L.); 3Program in Environmental and Polymer Engineering, Graduate School of Inha University, Incheon 22212, Republic of Korea; lsh1936@inha.edu (S.L.); alsdn2242@naver.com (M.B.); kmr1101@inha.edu (M.K.); yoototoasdf@naver.com (M.Y.); 4Particle Pollution Research and Management Center, Incheon 21999, Republic of Korea; 5Korea Dyeing & Finishing Technology Institute (DYETEC Institute), Daegu 41706, Republic of Korea

**Keywords:** evaporation-driven energy harvesting, electrospun nanofiber, support substrate, generation efficiency, nanocarbon black

## Abstract

Water evaporation-driven energy harvesting is an emerging mechanism for contributing to green energy production with low cost. Herein, we developed polyacrylonitrile (PAN) nanofiber-based evaporation-driven electricity generators (PEEGs) to confirm the feasibility of utilizing electrospun PAN nanofiber mats in an evaporation-driven energy harvesting system. However, PAN nanofiber mats require a support substrate to enhance its durability and stability when it is applied to an evaporation-driven energy generator, which could have additional effects on generation performance. Accordingly, various support substrates, including fiberglass, copper, stainless mesh, and fabric screen, were applied to PEEGs and examined to understand their potential impacts on electrical generation outputs. As a result, the PAN nanofiber mats were successfully converted to a hydrophilic material for an evaporation-driven generator by dip-coating them in nanocarbon black (NCB) solution. Furthermore, specific electrokinetic performance trends were investigated and the peak electricity outputs of *V_oc_* were recorded to be 150.8, 6.5, 2.4, and 215.9 mV, and *I_sc_* outputs were recorded to be 143.8, 60.5, 103.8, and 121.4 μA, from PEEGs with fiberglass, copper, stainless mesh, and fabric screen substrates, respectively. Therefore, the implications of this study would provide further perspectives on the developing evaporation-induced electricity devices based on nanofiber materials.

## 1. Introduction

Diverse renewable energy systems and their efficiency improvements have recently emerged as significant considerations in responding to climate change and promoting sustainability. Over the decades, various mechanisms and principles of energy harvesting technologies have been introduced and applied, such as solar, hydroelectric, and wind power generation systems. These advancements have enabled humanity to strive towards the promise of achieving a carbon-free (or net-zero) society. However, these conventional renewable energy systems (i.e., solar, hydroelectric, and wind power, etc.) are inevitably dependent on climate, weather, and geographical conditions as well as natural resource potentials. Several researchers have expressed concerns regarding the significant environmental and ecological impacts associated with renewable energy systems. These include air pollution emissions during the manufacturing processes [1,2,3]; social and wildlife disturbances, such as noise, bird collisions, and deforestation [4,5,6]; and issues related to waste (water) discharge and chemical misuse [7,8]. As a consequence, innovative alternatives are urgently required to overcome the critical limitations of conventional renewable energy systems.

Energy harvesting technologies utilizing water, such as water waves [9,10], water dropping [11,12], water evaporation [13,14,15,16], and ambient air humidity [17,18], are highly recommended options. Meanwhile, triboelectric and piezoelectric generators have been widely introduced as promising technologies for harvesting mechanical energy at the nanoscale. These technologies offer opportunities for the development of efficient and self-sustaining power sources for diverse applications [19,20,21,22]. In particular, evaporation-driven energy harvesting systems, emulated from a natural phenomenon in plants’ stomata (i.e., water evaporation from roots to leaves), have recently gained significant attention among material scientists and engineers due to their highly efficient electrokinetic effects. Evaporation-driven energy harvesting systems can convert environmentally low-grade energy potential into electricity through the physical and chemical interactions between water and materials. The main mechanisms for evaporation-driven power generation, such as streaming potential, pseudo-streaming, electron drag effect, and ion gradient diffusion, have been broadly investigated in several studies including the potential applications [23,24]. When water is intentionally injected or/and consumed from moist air into or by an evaporation-driven generator, electrokinetic phenomena are induced by the movement of charged ions through a liquid electrolyte flowing over the capillary channel [23,24]. Because of the capillary flow within the highly porous material, the wet side, where water is injected or/and consumed, undergoes a decrease in potential compared to the dry side. As a result, electrical current can flow from the wet side to dry side without being subject to specific conditions or environmental influences. Despite the technology being in the early stages of development, evaporation-driven electricity generation has been highlighted for its potential applications in sensors [25], small devices [14,26,27], and desalination [28].

Several functional materials including cellulose (or cotton), carbon materials, metal oxides, and polymers have been broadly demonstrated to generate electricity through the hydrovoltaic effect in various research fields. For instance, Zhang et al. [29] developed a highly efficient hydroelectric generator, achieving a maximum open-circuit voltage (*V_oc_*) and short-circuit current (*I_sc_*) of approximately 0.5 V and 1.3 mA, respectively, using a piece of carbon black-coated wood sponge (2  ×  4  ×  0.2 cm^3^). They also achieved a *V_oc_* of up to 33 V by simply integrating 60 units in series, which could directly drive commercial electronic devices. Wu et al. [30] emphasized the significance of a large surface area for excellent ion-exchange capacity by applying a honeycomb-structured reduced graphene oxide (rGO) film (1.5 × 2 cm^2^) that retains abundant interconnected microchannels. The authors further confirmed that an enhanced output power was observed from the honeycomb-structured rGO film with a *V_oc_* of ~0.53 V and *I_sc_* of ~1.5 μA cm^−2^, which featured abundant interconnected microchannels as opposed to the layer-structured rGO film with nanochannels with a *V_oc_* of ~0.078 V and *I_sc_* of ~0.5 μA cm^−2^ in DI water. Sun et al. [31] developed ceramic (SiO_2_) nanofibers to enhance the electrokinetic effects driven by water evaporation force. They were able to supply a stable voltage and current output of approximately 0.48 V and 0.37 μA from a single generator, and even successfully applied it in a commercial digital calculator with several connections in series and parallel. A 3D polypyrrole (PPy) was also introduced as a promising vapor-activated power generator, owing to its considerable advantages such as easy synthesis, relatively higher stability, and hydroscopic properties under moist environments [32]. More recently, Fang, et al. [33] and Dao et al. [34] have broadly reviewed emerging hydrovoltaic technologies based on carbon black and porous materials. These review papers investigated the overall mechanisms of evaporation-driven energy harvesting, previous research outcomes, factors influencing electricity output performance, and potential applications. Therefore, both professionals and beginners in evaporation-driven energy harvesting technologies could gain valuable knowledge and perspectives from these reviews.

Among the composite materials used for evaporation-driven energy harvesting, polymers, broadly applied in various fields, could be a beneficial source of generation devices due to their cost-effectiveness, lightweight nature, ease of functionalization, and comparable mechanical strength. Moreover, they can be formed into ultrafine fibers with diameters ranging from several hundred nanometers to micrometers using a simple electrospinning technique. As a result, electrospun nanofiber mats possess morphologically and geometrically notable characteristics, such as abundant nanochannels, higher levels of porosity, and increased vapor permeability. Meaningful attempts have been made to explore evaporation-driven generation outputs based on nanofiber mats with various types of metal electrodes [14,35], thicknesses, heights, and width variations [27,36], as well as nanofiber diameter distributions [31,37]. Despite this, a thin electrospun nanofiber mat physically and structurally requires a support substrate, e.g., mesh screens made of different materials, to maintain its shape and increase durability. Here, the choice of different types of support substrates for a nanofiber mat used as an evaporation-driven generation device is one of the influential factors affecting the efficiency of the electricity output. In this work, we report on the fabrication of a porous polyacrylonitrile (PAN) nanofiber mat using the electrospinning technique for developing an evaporation-driven electricity generator. Due to the physical and chemical characteristics of nanocarbon black powder, the surface of the PAN nanofiber mat was successfully transformed into a hydrophilic device, thereby creating PAN nanofiber-based evaporation-driven electricity generators (PEEGs) through dip-coating. Furthermore, the electricity generation outputs of PEEGs with various types of support substrates, including fiberglass, copper, stainless mesh, and fabric screen, were also extensively analyzed and compared.

## 2. Materials and Methods

### 2.1. Materials

Polyacrylonitrile (PAN, average MW = 150,000) and *N,N*-Dimethylformamide (DMF, ≥99.9%) were purchased from Sigma-Aldrich Co., Ltd., St. Louis, MO, USA. Nanocarbon black powder (NCB), used to increase the specific surface area of the PAN nanofiber mat, was obtained from Mitsubishi Chemical Co., Ltd., Tokyo, Japan. To uniformly disperse the NCB in deionized (DI) water, cetyl trimethyl ammonium bromide (CTAB), obtained from Tokyo Chemical Industry Co., Ltd., Tokyo, Japan, was used. Different substrates, including fiberglass, copper, stainless mesh, and fabric screen, used to support the thin PAN nanofiber mat with a thickness of several hundred micrometers, were purchased as commercial products. To fill the nanochannels with the ion solution for electricity generation, calcium chloride (CaCl_2_, ≥97.0%) powders were obtained from Sigma-Aldrich Co., Ltd. An adhesive copper foil tape for electrode connection of the generators was obtained from Teraoka Seisakusho Co., Ltd., Tokyo, Japan.

### 2.2. Fabrication of PAN Nanofiber Mat

A high-voltage power supply (LNC 20000-3neg, Heinzinger Electronic GmbH, Germany), syringe pump (SPLab01, DK Infusetek Co., Ltd., Shanghai, China), and speed control motor (K7IG15NC-SU, GGM Co., Ltd., Seoul, Republic of Korea) with a rotating drum collector were arranged within an acrylic ventilation chamber for the electrospinning process. For the electrospinning process, a homogeneous PAN solution with a concentration of 8 wt% was prepared by stirring in DMF at room temperature for 24 h. PAN is a general synthetic polymer that possesses polar nitrile (-CN) groups along its backbone, providing sites for chemical modification and functionalization. The electrospinning solution was then loaded into a syringe equipped with five 22 G stainless steel needles (with an inner diameter of 0.7 mm) arranged in a multi-nozzle configuration, which was mounted onto a syringe pump. Before operation, aluminum foil measuring 15 × 30 cm^2^ was wrapped around the collector to facilitate the deposition of nanofibers. Electrospinning was conducted with a voltage drop of 15 kV and a distance of 10 cm between the syringe needle tips and the collector. The solution extrusion rate from the syringe pump was set at 32.0 μL min^−1^ (6.4 μL min^−1^ per each needle) and the fabrication process was conducted at 23 °C and a relative humidity of 30%. The obtained PAN nanofiber mat was peeled off from the aluminum foil and affixed onto different types of support substrates, i.e., fiberglass, copper, stainless mesh, and fabric screen (with a thickness of approximately 1.0 mm).

### 2.3. Characterization of Evaporation-Driven Electricity Generator

Incorporating carbon–polymer materials into a composite offers a highly flexible approach for modulating material properties, enhancing productivity and deducing costs. NCB is a form of elemental carbon characterized by its fine particle size and high surface area, possessing a highly porous structure. First, 0.25 g of NCB was dispersed in 40 mL of DI water with 0.6 g of CTAB surfactant. Then, the NCB solution was mixed via sonication for 1 h to achieve uniform dispersion. To prepare PEEGs, PAN nanofiber mats fixed on different types of support substrates were dip-coated in the NCB solution and then heated in a 90 °C oven (SH Scientific Co., Ltd., Sejong, Republic of Korea) for 1 h. Afterward, two pieces of adhesive copper foil tape (5.0 × 60.0 mm^2^) were folded in half and connected to each end of the NCB-coated PAN nanofiber mat, which was fixed onto different substrates as the electrodes. This approach, connecting electrodes to the water evaporation-driven energy generators, was previously introduced by Tabrizizadeh et al. [35]. Finally, PEEGs with different support substrates were prepared. To analyze the characterizations of PGGE, scanning field emission electron microscopy (SEM, JSM-7100F, JEOL Co., Ltd., Tokyo, Japan), energy-dispersive X-ray spectroscopy (EDS, JSM-7100F, JEOL Co., Ltd., Japan), and Fourier-transform infrared spectroscopy (FT-IR, Spectrum Two, PerkinElmer Inc., Waltham, MA, USA) were used. The generation performances, including open-circuit voltage (*V_oc_*) and short-circuit current (*I_sc_*) curves, were measured using a Keithley 2400 source meter (Keithley Instruments, Inc., Cleveland, OH, USA).

## 3. Results and Discussion

### 3.1. Preparation and Characterization of PAN Nanofiber Mat

The electrospun nanofiber mat was fabricated using the horizontal configuration of the conventional electrospinning technique. PAN, widely employed in electrospinning, was chosen as the polymer for preparing the electrospun nanofiber mat, and the fabrication process followed the procedure outlined in Figure 1a. The surface of the nanofiber mat was examined to assess its uniform distribution and disordered fibrous structure, with fiber diameters ranging from approximately 200 to 250 nm, using SEM imaging (Figure 1b). From Figure 1c, it is evident in the cross-sectional view that the nanofibers were densely packed, forming a structured mat characterized by hierarchically and irregularly interlaced nanofibers. The fabricated PAN nanofiber mat exhibited excellent flexibility and significant softness (Figure 1d). Furthermore, the PAN nanofiber mat exhibited favorable characteristics of shear processability and mechanical stability. The PAN nanofiber mat, weighing 0.045 g and measuring 20.0 × 40.0 × 0.8 mm^3^ without a supporting substrate, demonstrated the ability to withstand a weight of 250 g, as illustrated in Figure 1e. Despite the stable durability of a PAN nanofiber mat in dry conditions, its ability to retain shape may not be guaranteed when it is utilized in evaporation-driven generation devices and fully covered with moisture. To prevent distortion of the thin PAN nanofiber mat, four different support substrates, i.e., fiberglass, copper, stainless mesh, and fabric screen, were affixed together. Subsequently, the PAN nanofiber mats with different support substrates were cut into rectangle shapes measuring 2.0 × 4.0 cm^2^.

### 3.2. Characterization of PEEGs

In the evaporation-driven generation mechanism, NCB plays a significant role in enhancing the hydrophilic property within nanochannels. Leveraging the abundant tubular capillary structure and high surface-area-to-volume ratio of the PAN nanofiber mat, NCB allows for better water absorption and transport capabilities when collaborating together. To prepare PEEGs, the PAN nanofiber mats with different support substrates were dip-coated in the NCB solution in a Petri dish for 10 s. Afterward, the coated nanofiber mats were placed on a glass plate and transferred to an oven for drying. Additionally, the weight of the PAN nanofiber mat after NCB dip-coating was about 0.059 g, indicating that approximately 0.014 g of NCB powders were coated on each individual PEEG.

The distinctive features of each PEEG were verified through SEM images and EDS patterns. The uniform distribution of NCB on the nanofiber mat was readily apparent as shown in Figure 2a, which presents the nano-sized carbon black particles aggregated on the surfaces of PAN nanofibers. Thus, they could offer a larger specific surface area than the non-coated state and also improve the physical characteristics for the water evaporation-driven energy harvesting application. This was made more distinct by the elements of nitrogen (N, green), bromine (Br, teal) and carbon (C, red), homogeneously dispersed in the PAN nanofiber matrix at the nanometer level. Specifically, the clear distribution of the N element most likely followed the irregular arrangement of nanofibers, because PAN and DMF were the main sources of the electrospinning solution (Figure 2b). There were more pronounced clusters observed in the individual EDS maps of the Br and C elements compared to the N element as shown in Figure 2c and Figure 2d, respectively. After fabricating the electrospun PAN nanofiber mat, it was dip-coated into the NCB solution composed with CTAB surfactant. Hence, both the Br and C elements distributed on the nanofiber surfaces were strongly detected in specific regions, which is in agreement with the PAN nanofibers and NCB particles in the SEM images (Figure 2a).

The modification of PAN nanofiber mats was also examined using FT-IR. As depicted in Figure 2e, absorption peaks at 2243 cm^−1^ corresponding to the –C≡N were observed in both cases of non-coated and NCB-coated PAN nanofiber mats. Compared to the spectrum of the non-coated PAN nanofiber mat, new peaks at 1665 cm^−1^ (C=C stretch) and 875 cm^−1^ (C=O bend) were observed on the spectrum of the NCB-coated PAN nanofiber mat. Additionally, absorption peaks at 3660 cm^−1^, 2918 cm^−1^, and 1070 cm^−1^, caused by the presence of O–H, –C–H stretch, and C–O stretch, respectively, were significantly increased. Remarkably, the peaks at 3660 cm^−1^ and 2008 cm^−1^ on the spectrum of the NCB-coated PAN nanofiber mat were, respectively, attributed to the hydroxyl (O–H) and carboxyl (C=O) groups of –COOH, confirming that the surface of the PEEG was successfully modified to exhibit a hydrophilic property. Eventually, the PEEGs were obtained from the NCB-coated PAN nanofiber mats with different support substrates (Figure 2f) by connecting two pieces of adhesive copper foil tape on each end side, serving as electrodes as shown in Figure 2g.

The hydrophobic surface of the PAN nanofiber mat was completely altered to possess a hydrophilic nature, ensuring a continuous and sufficient water supply for evaporation-driven generation. Figure 3a describes how the nanofiber mat with micropores facilitates the flow of absorbed water along the disorderly intertwined non-coated and NCB-coated nanofibers, driven by capillary force. However, the different water transport velocities can be observed due to the surface modification of PAN nanofibers from–C≡N to –COOH through the carbon coating process, as discussed earlier. To directly compare the wettability between the raw (i.e., non-coated) PAN nanofiber mat and the PEEG, water transporting velocities in the vertical direction were examined using the same mat size, support substrate (fabric screen), and experimental conditions (e.g., room temperature, humidity, and the same liquid (DI water) provided). When water was absorbed at the bottom of the raw PAN nanofiber mat, it ascended slowly to the opposite side, stopping at a third of the way from the starting point in about 45 s (Figure 3c). In contrast, the directional movement of water in the PEEG was more clearly evident, demonstrating its super-hydrophilic characteristic, as shown in Figure 3d. The PEEG immediately absorbed the liquid water upon contact with the water surface, swiftly transporting it to the top (approximately 40 mm) of the NCB-coated nanofiber mat in about 1 min. The micro- and nanochannels in the PAN nanofiber mat can accelerate the water wetting velocity, and the even distribution of coated NCB with hydrophilic groups on the nanofiber surface provides an additional effect on mass transfer. The combined influence of the above factors eventually resulted in PEEGs having excellent water delivery ability. Meanwhile, both the non-coated PAN nanofiber mat and PEEG showed gradually decreasing water transport velocities as the height increased (Figure 3b).

### 3.3. Evaluation of Electrical Performances by Different Support Substrates

In this study, the electrical performance of PEEGs with different support substrates was evaluated to understand the potential effect of support substrate types on evaporation-driven electricity outputs. Following the general mechanism and operating principle, the bottom electrode of the PEEG was immersed in a liquid. In this study, a 3.3 M calcium chloride (CaCl_2_) solution, a typical deliquescent chemical, was used instead of DI water. The CaCl_2_ solution provides additional ions, which play a role in increasing the charge density on the carbon surface and reducing the resistance of the generator [13]. Furthermore, CaCl_2_ can absorb moisture from the surrounding atmosphere and continuously provide water to the generator. Therefore, the PEEG was expected to maintain stable wetting asymmetry as well as autonomous electrical generation through artificial aqueous circulations. In fact, other salt solutions, such as KCl, NaCl, and LiCl, have also been widely used in evaporation-driven electricity generation [13,14,38]. However, controversial questions still arise, necessitating further investigations. This is because the electrokinetic performance of evaporation-based systems varies depending on factors such as the material type of the generation devices, their size and shape, and the surface treatment methods employed.

Figure 4a,b illustrate the maximum electricity outputs in terms of voltage (mV) and current (μA), respectively, from PEEGs with different support substrates (fiberglass, copper, stainless mesh, and fabric screen). The PEEG with the fabric screen substrate (PEEG*_Fabric_*) showed the highest *V_oc_* measuring 215.9 mV, while values of 150.8 mV, 6.5 mV, and 2.4 mV were observed from PEEG*_Fiberglass_*, PEEG*_Copper_*, and PEEG*_Stainless_*, respectively. The maximum *I_sc_* from PEEG*_Fiberglass_* demonstrated the highest current performance, measuring 143.8 μA among the different support substrates, compared to the maximum *I_sc_* of PEEG*_Fabric_*, PEEG*_Copper_*, and PEEG*_Stainless_*, which were 121.4 μA, 60.5 μA, and 103.8 μA, respectively. For the comprehensive understanding of PEEG performance, the maximum *V_oc_* and *I_sc_* were analyzed under varied load resistances, as summarized in Figure 4c. When the load resistances increased from 1 Ω to 10 Ω, the *V_oc_* exhibited a noticeably low level (≤1.9 mV), and the *I_sc_* was maintained at approximately 58.1 μA. As the load resistance increased from 100 Ω to 10 kΩ, the voltage rose from 3.6 mV to nearly 270 mV. Meanwhile, the *I_sc_* gradually increased from 57.9 μA to 217.5 μA as the load resistance further increased from 10 Ω to 1 kΩ, then decreased to 42.0 μA as the resistance further increased from 1 kΩ to 10 kΩ. The peak output power from PEEGs reached a maximum value of 36.1 nW with a load resistance of approximately 1 kΩ.

To assess the long-term stability of the PEEGs with different support substrates, the electricity outputs were monitored until they reached a steady state of power generation efficiency at 23 °C and a relative humidity of 30% (Figure 4d–g). When the CaCl_2_ solution was dropped onto the PEEGs, the generation outputs of *V_oc_* and *I_sc_* quickly increased owing to the maximum wet differences between two electrodes: the area where moisture was injected and the opposite side of the PEEG. Following the peak electricity outputs, both *V_oc_* and *I_sc_* gradually decreased and eventually stabilized when the PEEG was fully wetted, maintaining a continuous circulation of water supply and evaporation. Although the same PAN nanofiber mats were used, the electricity output performances after the stabilization of generation exhibited different trends depending on their support substrate types. For example, PEEG*_Fiberglass_* and PEEG*_Fabric_* showed a significant electrokinetic *V_oc_* of 101.2 mV and 91.1 mV along with an *I_sc_* of 81.8 μA and 72.1 μA, respectively. However, PEEG*_Copper_* and PEEG*_Stainless_* exhibited the lower *V_oc_* of 2.3 mV and 1.8 mV and *I_sc_* of 32.0 μA and 49.3 μA, respectively, compared to the fiberglass mesh and fabric screen support substrates. In particularly, the electricity output of voltage and current from PEEG*_Copper_* was rather erratic and unstable, rendering it unsuitable for future energy harvesting devices.

These experimental output performances were obtained under a normal environmental state for generating evaporation-driven electricity. Several formal studies have reported that the generation efficiencies of evaporation-driven power devices are highly dependent on the relative humidity of the surrounding environment. According to Yun et al. [38], as the relative humidity increased from 45% to 85%, the voltage and current generated gradually declined from about 0.4 V to nearly 0 V and from 0.18 μA to 0.02 μA, respectively. This phenomenon was also thoroughly investigated by Lv et al. [27]. They found that the highest output voltage occurred at a relative humidity of 30%, and then it continuously decreased with an increase in relative humidity from 30% to 75%. Furthermore, the device exhibited low output voltage when placed under relative humidity levels of 80% and above.

This study primarily focuses on comparing the generation output performance of PEEGs depending on different types of support substrates. Despite the exploration of better power generation performance, as summarized in Table 1, this study remains meaningful in providing further perspectives on developing evaporation-driven electricity generation devices using electrospun nanofiber mats.

## 4. Conclusions

In this study, an electrospun PAN nanofiber mat was developed as an evaporation-driven electricity generator, and its electricity output performance was examined with various types of support substrates. Initially, it was confirmed that the hydrophobic nature of the PAN nanofiber mat surface (–C≡N) was effectively transformed into a highly moisture-absorbent material (–COOH), enabling sustainable water supply and evaporation following dip-coating with NCB. The PAN nanofiber-based evaporation-driven electricity generators (PEEGs, 20.0 × 40.0 × 0.8 mm^3^) utilizing different support substrates showed distinct electrokinetic performance trends. Peak electricity outputs were recorded, with *V_oc_* values of 150.8 mV, 6.5 mV, 2.4 mV, and 215.9 mV and *I_sc_* values of 143.8 μA, 60.5 μA, 103.8 μA, and 121.4 μA shown by the PEEGs constructed with fiberglass, copper, stainless mesh, and fabric screen substrates, respectively. The electrokinetic performances of the PEEGs were notably influenced by the different types of support substrates, mostly due to their different load resistances. The maximum power density of 36.1 nW was reached with a load resistance of approximately 1 kΩ under ambient conditions (23 °C, 30% RH) using a 3.3 M CaCl_2_ solution. In particular, the types of support substrates of PEEGs were also further confirmed to influence the consistency of electricity generation. Specifically, the *V_oc_* and *I_sc_* from PEEG*_Fiberglass_* and PEEG*_Fabric_* exhibited greater stability compared to those from PEEG*_Copper_* and PEEG*_Stainless_*. This could be because the conductivity of the supporting substrate affects the operational characteristics of the device. Last but not least, this study firstly assessed the potential impacts of different support substrates on electricity generation output, focusing on PAN as a case study. Such insights could provide valuable information for the development and improvement of evaporation-induced energy-harvesting devices utilizing nanofibers in the future.

## Figures and Tables

**Figure 1 polymers-16-01180-f001:**
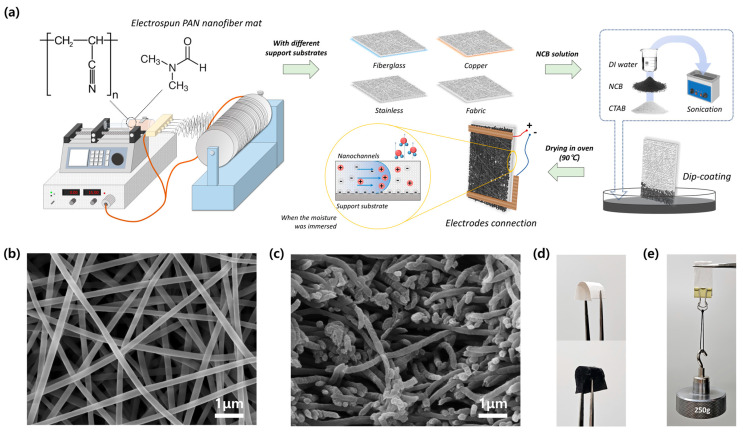
The preparation and morphological characterization of PAN nanofiber mats. (**a**) Schematic diagram of electrospinning fabrication by the horizontal setting for developing PEEGs; (**b**) the surface of the fabricated PAN electrospun nanofiber mat; (**c**) cross-sectional cutting view of nanofiber structure with nanopores distributed; (**d**) flexibility and softness of non-coated and NCB-coated PAN nanofiber mats; (**e**) PAN nanofiber mat without support substrate against a dragging weight of 250 g.

**Figure 2 polymers-16-01180-f002:**
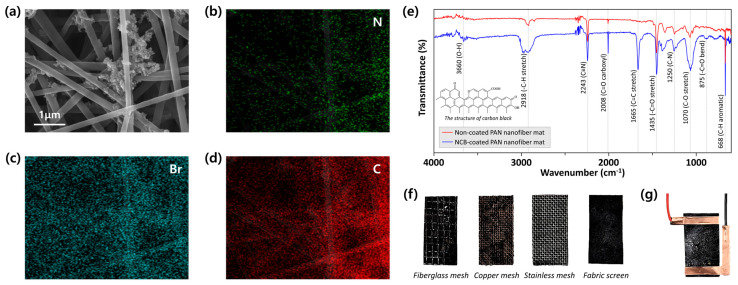
Characterization of PEEGs. (**a**) SEM image of NCB-coated PAN nanofibers and its EDS maps of (**b**) N (green), (**c**) Br (teal), and (**d**) C (red) elements; (**e**) FT-IR spectroscopy to prove the property modification of the PAN nanofiber mat; (**f**) back sides of NCB-coated PAN nanofiber mats with different support substrates; (**g**) PEEGs with copper electrodes connected with single-end electrical alligator clips.

**Figure 3 polymers-16-01180-f003:**
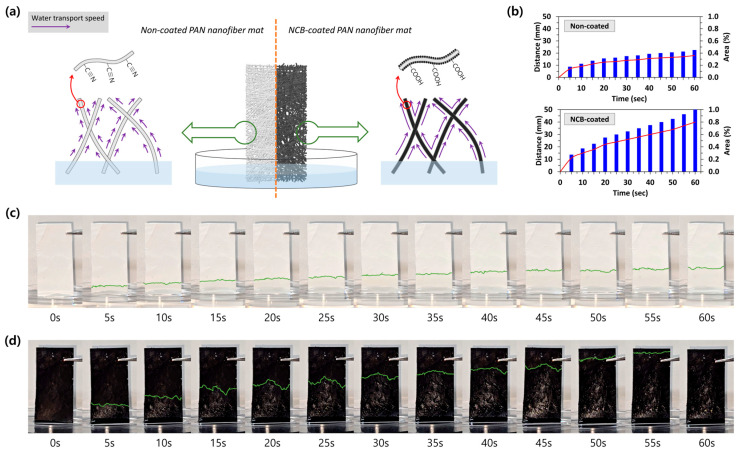
Comparison of the water transfer abilities between the raw PAN nanofiber mat and the PEEG with the fabric substrate. (**a**) Schematic illustration of water transports in non-coated and NCB-coated PAN nanofiber mats; (**b**) distance and wetted area changes of raw and NCB-coated PAN nanofiber mats transporting water in vertical direction; distance changes of (**c**) non-coated PAN nanofiber mat and (**d**) NCB-coated PAN nanofiber mat transporting liquid water in vertical direction.

**Figure 4 polymers-16-01180-f004:**
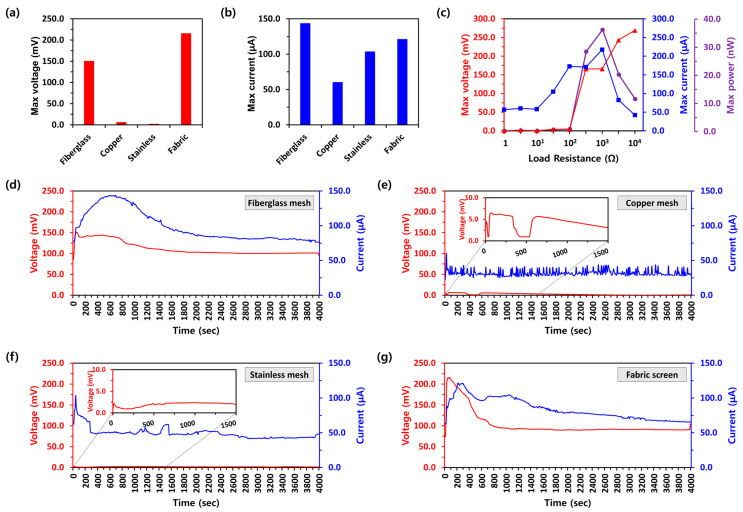
The maximum power generation performances of (**a**) voltages and (**b**) currents from PEEGs according to different types of support substrate. (**c**) Maximum output voltage, current, and power of the PEEG with load resistance variations. The long-time stability of PEEGs with the support substrates (**d**) fiberglass mesh; (**e**) copper mesh; (**f**) stainless mesh; (**g**) fabric screen.

**Table 1 polymers-16-01180-t001:** Summary of the generation output performances from previous studies.

Materials	Device Size	Generation Outputs		Years	Ref
		*V_oc_*	*I_sc_*		
Carbon nanotube film	0.5 × 1.0 cm^2^	0.98 V	515.6 μA	2023	[39]
Ceramic (SiO_2_) nanofiber	2.0 × 6.0 cm^2^	0.48 V	0.37 μA	2021	[31]
Polymer foam	15.0 × 15.0 × 10.0 mm^3^	0.25 V	42.0 μA	2021	[40]
Natural wood	5.0 × 5.0 × 1.0 cm^3^	0.3 V	10.0 μA	2020	[41]
Lower-grade gums	1.0 × 2.5 cm^2^	0.4 V	3.0 μA	2020	[42]
CuO nanowire films	1.5 × 6.0 cm^2^	0.45 V	0.23 μA	2020	[43]
Flexible hydroelectric films	16.0 × 4.0 cm^2^	2.5 V	0.8 μA	2019	[44]
Cotton fabric	30 × 90 × 0.12 mm^3^	0.74 V	22.5 μA	2019	[13]

## Data Availability

Data are contained within the article.

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
