# Peer review of "Evaporation-Driven Energy Generation Using an Electrospun Polyacrylonitrile Nanofiber Mat with Different Support Substrates"

_polymers, 2024, doi:10.3390/polym16091180_

Round 1

Reviewer 1 Report

Comments and Suggestions for Authors

In this paper (polymers-2947106), the authors reported an evaporation-driven generator based on electrospun polyacrylonitrile nanofiber mat, and the results are acceptable. But there are some problems in the writing, motivation, presentation, and discussion of the results. Major modifications are necessary.

1.        Title: The first letters of all words (except for articles and prepositions) need to be capitalized.

2.        Title and results: “Evaporation”: From the results (Fig. 3a), it seems that the authors did not use water vapor or moisture, but rather liquid water. This statement seems inaccurate.

3.        Introduction: (1) “…solar, hydroelectric, and wind…”. Widely reported triboelectric/ piezoelectric nanogenerators need to be included. In addition, new electrochemical self-powered humidity sensors related to moisture can also be included. These statements require literature as support. (2) The research motivation is unclear. What is the application background of evaporation-driven generator? Previously reported ion gradient generators were commonly used for energy supply, storage, or self-powered humidity detection. Suggest a comprehensive and thorough discussion. (3) The research status of electrospun nanofiber mat for moisture power generation and sensing is not fully discussed, such as Chem. Eng. J., 2022, 438, 135588.

4.        P153: mL instead of ml.

5.        “The CaCl2 solution provides additional protons,…”. How to provide protons? The reviewer conservatively believes that Ca and Cl ions are involved in conduction rather than primarily protons. Why not use other salts, such as NaCl and LiCl. It seems that more mobile conductive ions can be generated due to their lightweight.

6.        “…23℃ and a relative humidity of 30% (Figs. 4d-4g)…”. (How was 30% RH obtained.) (2) How does RH affect the power generation performance of device? Suggest providing output voltage and power generation characteristics at different RH levels.

7.        In terms of power generation performances, it is recommended to compare it with previous reports to highlight some of the advantages of this article.

8.        Lack of detailed mechanism analysis.

9.        References: Some reference information is missing, such as [25]

Comments on the Quality of English Language

Minor editing of English language required.

Author Response

We would like to express our sincere gratitude for the valuable feedback provided by Reviewer #1 on our manuscript titled "Evaporation-driven Energy Generation Through Electrospun Polyacrylonitrile Nanofiber Mat with Different Support Substrates," submitted to Polymers. Your insightful comments and suggestions have been instrumental in improving the quality and clarity of our work.

In response to Reviewer #1's comments, we have made significant revisions to the manuscript. Details of the major changes are provided in the attached document.

Reviewer 2 Report

Comments and Suggestions for Authors

This work deals with the application of modified PAN nanofibers for evaporation- driven electricity generators and the effect of the different supporting substrates on the efficiency of the device. The idea is quite interesting and the authors have done an excellent experimental work.

I would only like to give one or two recommendations to further improve the manuscript:

- You can calculate the apparent density of the PAN nanofiber mat and compare it with the nominal polymer density to find the void volume.

- You can also quantifically calculate the amount of the NCB load from the weight increase - and correlate with the EDS.

- It seems that the conductivity of the supporting substrate affects the operational characteristics of the device-  the erratic behaviour of the current-time curves is probably sign of partial discharge. 

I also found some typos:

- Page 2, line 93: "These" instead of "these"

- Page 5, line 213: "compared" instead of "compare"

- Page 5, lines 221-222: it is better to use "spectrum" instead of "curves"

- Page 6, line 243: it is better to use "velocities" instead of "speeds"

- Page 7, line 277: it is better to use "aqueous" instead of "hydrological"

All in all, the paper can be accepted for publication after these minor revisions.

Author Response

We would like to express our sincere gratitude for the valuable feedback provided by Reviewer #2 on our manuscript titled "Evaporation-driven Energy Generation Through Electrospun Polyacrylonitrile Nanofiber Mat with Different Support Substrates," submitted to Polymers. Your insightful comments and suggestions have been instrumental in improving the quality and clarity of our work.

In response to Reviewer #2's comments, we have made significant revisions to the manuscript. Details of the major changes are provided in the attached document.

Reviewer 3 Report

Comments and Suggestions for Authors

In this study, as an approach of energy harvesting technology, authors have developed a polyacrylonitrile nanofiber mat for the purpose of generating evaporation-induced electricity by utilization of different support substrates including fiberglass, copper, stainless 28 mesh, and fabric screen. The surface of the mat was hydrophlized by nanocarbon black powder using dip coating method. The entrance of moisture into capillary channel in porous mat leads to movement of charged ion, inducing electricity.

The manuscript is valuable and written well. However, authors should address the issues.

·         Authors should explain more about polyacrylonitrile polymer and nanocarbon black powder.

·         How authors ensure that there is not interfering moisture in mat after dip-coating in the NCB solution and then heating in a 90℃ oven for 1 hour.

·         Authors should report the thickness of mat after and before dip-coating with and without supports. Furthermore, swelling ratio of the mat is important and should be reported.  

·         Authors should prove the hydrophilicity of the mats upon NCB treatment by measuring the contact angle of the fibers and discuss about it.

·         Mechanical properties of the mat and devices have not been mentioned.

·         Authors should compare their finding with the similar studies.

Best

Comments on the Quality of English Language

Minor editing of English language required.

Author Response

We would like to express our sincere gratitude for the valuable feedback provided by Reviewer #3 on our manuscript titled "Evaporation-driven Energy Generation Through Electrospun Polyacrylonitrile Nanofiber Mat with Different Support Substrates," submitted to Polymers. Your insightful comments and suggestions have been instrumental in improving the quality and clarity of our work.

In response to Reviewer #3's comments, we have made significant revisions to the manuscript. Details of the major changes are provided in the attached document.

Round 2

Reviewer 1 Report

Comments and Suggestions for Authors

The response and revised manuscript are satisfactory, and it is recommended to accept.

In the proof stage, check the format of the references, such as using abbreviations for journal names and subscripts for numbers in chemical formulas.

Reviewer 3 Report

Comments and Suggestions for Authors

.

Comments on the Quality of English Language

 Minor editing of English language required